# Drinking Water Supplemented with Acidifiers Improves the Growth Performance of Weaned Pigs and Potentially Regulates Antioxidant Capacity, Immunity, and Gastrointestinal Microbiota Diversity

**DOI:** 10.3390/antiox11050809

**Published:** 2022-04-21

**Authors:** Qing-Lei Xu, Chang Liu, Xiao-Jian Mo, Meng Chen, Xian-Le Zhao, Ming-Zheng Liu, Shu-Bai Wang, Bo Zhou, Cheng-Xin Zhao

**Affiliations:** 1College of Animal Science and Technology, Nanjing Agricultural University, Nanjing 210095, China; 2019205004@njau.edu.cn (Q.-L.X.); 2019805121@njau.edu.cn (C.L.); 2020805124@stu.njau.edu.cn (M.C.); 2020805123@stu.njau.edu.cn (X.-L.Z.); 2020205018@stu.njau.edu.cn (M.-Z.L.); 2Yantai Jinhai Pharmaceutical Co., Ltd., Yantai 265323, China; moxiaojian@jinhaiyaoye.com; 3College of Animal Science and Technology, Qingdao Agricultural University, Qingdao 266109, China; wangshubai@qau.edu.cn

**Keywords:** weaned pigs, acidifier, antioxidants, animal feed, intestinal flora

## Abstract

This study evaluated the potential effects of adding acidifiers to the drinking water on the growth performance, complete blood count, antioxidant indicators, and diversity of gastrointestinal microbiota for weaned pigs. A total of 400 weaned pigs were randomly divided into four treatments. Pigs were fed the same basal diet and given either water (no acidifier was added, control) or water plus blends of different formulas of acidifiers (acidifier A1, A2, or A3) for 35 days. On d 18 and 35 of the experimental period, 64 pigs (four pigs per pen) were randomly selected to collect blood for a CBC test (*n* = 128) and an antioxidant indicators test (*n* = 128); 24 pigs (six pigs per group) were randomly selected to collect fresh feces (*n* = 48) from the rectum for 16S rRNA gene sequencing. Compared to the control, supplementing the drinking water with acidifiers improved the growth performance and survival rate of weaned pigs. Acidifier groups also increased serum catalase (CAT) and total antioxidant capacity (T-AOC) activities, while also displaying a decreased malondialdehyde (MDA) concentration compared to the control. The relative abundance of *Firmicutes* in the acidifier A1 group was greater than that in the control group (*p* < 0.05) on d 35; the relative abundance of *Lactobacillus* in the acidifier A1 group was greater than that in the control group (*p* < 0.05) on d 18 and 35. The microbial species *Subdoligranulum* or *Ruminococcaceae_UCG-005* had significantly positive correlations with ADG and ADFI or with serum antioxidant indicators, respectively. These findings suggest that supplementing the drinking water with an acidifier has a potential as an antioxidant, which was reflected in the improvement of growth performance, immunity, antioxidant capacity, and intestinal flora.

## 1. Introduction

Segregated early weaning (SEW) improved the fertility of sows and reduced the transmission of diseases from sows to piglets, which improved the production efficiency and provide economic benefits to the pig industry [1,2]. However, due to immature digestive organs, insufficient secretion of gastric acid and digestive enzymes, and low immunity of early weaned pigs, they often display early weaning stress syndrome, which is characterized by digestive dysfunction, growth retardation, and diarrhea [3,4]. After weaning, pigs are vulnerable to the invasion of pathogens, resulting in a dysfunction of the intestinal flora and immune system due to the changes of environment, feed, and other factors [5].

In the past, antibiotics had played an important role in the prevention and treatment of a pig’s diseases and the reduction of weaning stress [6,7]. However, long-term abuse of antibiotics affected the diversity of intestinal microbiota in newborn piglets and beneficial bacterial colonization, which led to subsequent intestinal diseases [8,9,10]. In addition, long-term abuse of antibiotics not only produces antibiotic-resistant bacteria, but also causes antibiotic residues in animal-derived food products and the environment [11,12]. Following the ban of antibiotic growth promoters, new alternatives are required to solve these problems.

As non-toxic, pollution-free, and non-resistance functional feed additives, acidifiers have played important roles in the improvement of animal growth, immunity, and intestinal health [13]. Supplementation with acidifiers improved the activity of pepsin and enhanced the digestion and absorption of nutrients in the gastrointestinal tract of weaned pigs [14]. In addition, acidifiers increased feed intake by inducing and stimulating taste buds, thus promoting the growth performance of pigs [8,15]. Acidifiers also improved the antioxidant capacity of piglets [16,17] and reduced the diarrhea caused by weaning stress [18]. Acidifiers decreased the pH value of the gastrointestinal tract [19], changed the composition of intestinal microorganisms, and inhibited the harmful microorganisms which are sensitive to low pH (such as *Enterobacteriaceae*) [20]. Therefore, acidifiers are increasingly being added to feed as an antibiotic-free growth promoter to replace antibiotics.

Previous studies investigated acidifiers as feed additives [21,22]. However, when the feed contains more protein or minerals with high buffer capacity, the effect of acidifiers may be weakened. At the same time, the acidifier was neutralized by the alkaline substances in the feed, thus losing the acidification effect [23]. The liquid acidifier can be added to the drinking water through the dosing device of the drinking water pipe, which is more convenient to add when needed. Moreover, acidified drinking water had the same effect as adding it to the feed [24,25].

Common drinking water compound organic acidifiers contained formic acid, acetic acid, and propionic acid [26]. As an important component of an organic acidifier, lactic acid decreased intestinal *Salmonella* and improved intestinal health [27]. Stevioside was widely used as a sweetener in feed additives to improve the feed intake of livestock and poultry [28,29]. Therefore, the compound organic acidifier containing formic acid, acetic acid, and propionic acid was selected as one treatment. On this basis, lactic acid was added as another treatment. In order to improve the taste of the drinking water, stevioside was added as the third treatment, and water without any additions was set as the control group. The effectiveness of acidified drinking water, containing a large number of organic acids, is largely based on the pH being lowered to a level of 4.0, at which *Enterobacteriaceae* cannot multiply [30]. Therefore, the acidified drinking water of each treatment group contains different proportions of organic acids, so as to adjust the pH value to 4.0. The present experiment aimed to evaluate the effects of acidified drinking water on the growth performance, immunity parameters, antioxidant capacity, and intestinal microflora of weaned pigs, and then compare which types of acidifiers are more suitable to be added to the drinking water of weaned pigs as an antibiotic substitute.

## 2. Materials and Methods

### 2.1. Pigs, Experimental Design, and Housing

This study was approved by the Nanjing Agricultural University Animal Care and Use Committee (SYXK Su 2017-0027). In this experiment, a total of 400 weaned pigs at 28 d of age were randomly divided into four groups: (1) acidifiers A1 group (continuous supply of acidified drinking water with 19% formic acid +19% acetic acid +3.5% propionic acid +15% lactic acid, pH = 4); (2) acidifiers A2 group (continuous supply of acidified drinking water with 22% formic acid +16% acetic acid +4% propionic acid, pH = 4); (3) acidifiers A3 group (continuous supply of acidified drinking water with 22% formic acid +16% acetic acid +4% propionic acid +0.5% stevioside, pH = 4); (4) control group (continuous supply of non-treated water, pH = 7). There were four replicates (pens) in each group, and 25 pigs per pen; the acidifiers were used for 8 h every day and the pigs were free to drink. A ratio of acidifier: water = 1:500 was used in this study. 

All pens were equipped with slatted floors, stainless-steel vibrating feeders, and cup drinking bowls. The temperature was automatically controlled between 22 to 27 °C by air-exhaust fans, evaporative cooling pads, and hot blast heaters. A water-powered proportional dosing pump (DOSATRON, INTERNATIONAL, Bordeaux, France) was used to supplement the liquid acidifiers into the drinking water for 35 d before the end of the nursery period. According to the standard of NRC (2012), a corn-soybean meal basal diet was formulated to meet the nutritional needs of pigs (Appendix A). Feed and water were provided ad libitum. The experimental pigs were immunized according to the specified immunization procedure.

### 2.2. Growth Performance and Diarrhea Rate

Pigs were weighed at d 0 and 35 of the experiment and the data was used to calculate the average daily gain (ADG). Average daily feed intake (ADFI) was recorded to calculate feed-to-gain ratio (F:G). The survival rates of pig were determined at the end of the experiment.

The diarrhea scores of pigs were recorded from 08:00 to 09:00 every day according to the criteria: 1-solid, well-formed feces; 2-loose and shapeless feces; 3-runny feces; and 4-watery diarrhea [31]. Diarrheal symptoms and mortality, if any, were recorded daily for each labelled pig during the trial period. The percentage of diarrhea occurrence in the total number of pigs in a pen (diarrhea severity), diarrhea duration (d), and feces score were calculated.

### 2.3. Complete Blood Count (CBC) Test

On d 18 and 35 of the experimental period, 64 pigs (four pigs per pen) were randomly selected to collect 5 mL of blood via jugular venipuncture. The blood was then divided equally into two parts: one for a CBC test (*n* = 128) and the other for the determination of antioxidant indicators (*n* = 128). A total of 23 CBC indicators of the whole blood samples were determined using animal specialty automatic physiological analysis instruments (Mindray, BC-5000 Vet, Shenzhen Mindray Biomedical Electronics Co., Ltd., Shenzhen, China) within several hours. The serum samples were obtained by centrifuging (4000× *g* for 10 min) at 4 °C, and stored at −80 °C until analysis.

### 2.4. Antioxidant Indicators Measurements

All the serum samples (*n* = 128) were used for antioxidant activity analysis. Antioxidant indicators, such as malondialdehyde (MDA), total superoxide dismutase (SOD), glutathione (GSH), glutathione peroxidase (GSH-Px), catalase (CAT), and total antioxidant capacity (T-AOC), were determined using ELISA Kits (Shanghai Langdun Biotechnology Co., Ltd., Shanghai, China) with assay sensitivities of 99.0% according to the manufacturer’s instructions. Briefly, we use a competitive method to detect the content of MDA, SOD, GSH, GSH-Px, and CAT in the sample. The samples were added to the ELISA plates that were pre-coated with antibodies, then biotin-labeled antigens were added, and the plates were incubated at 37 °C for 30 min. The two compete with antibodies to form immune complexes. The unbound biotin-labeled antigens are removed by rinsing in PBS with 0.15% Tween 20 (PBST), then horseradish peroxidase avidin (Avidin-HRP) is added, and the plates are incubated at 37 °C for 30 min. Avidin-HRP binds to biotin-labeled antigens, and, after washing, the bound HRP enzyme catalyzes tetramethylbenzidine (TMB) into a blue dye and then into yellow under the action of acid. We measured the absorbance (OD value) of each well at a wavelength of 450 nm within 10 min and calculated the test samples according to the standard curve.

T-AOC was measured using a ferric reducing ability of power (FRAP) assay. Briefly, antioxidants react with the ferric tripyridyl triazine (Fe^3+^-TPTZ) complex and convert it into ferrous tripyridyl triazine (Fe^2+^-TPTZ) under acidic conditions. The absorbance of T-AOC was measured at 593 nm. The reaction absorbance for MDA, SOD, GSH, GSH-Px, CAT, and T-AOC were measured using a microplate reader (Tecan, Austria GmbH, Grödig, Austria). The sensitivities of MDA, SOD, GSH, GSH-Px, and CAT were 0.15 nmol/mL, 0.1 ng/mL, 10 μg/mL, 0.3 ng/mL, and 0.2 ng/mL, respectively. Each sample was tested three times. The intra- and inter-assay coefficients of variation were less than 10%.

### 2.5. DNA Extraction and 16S rRNA Gene Sequencing

On d 18 and 35 of the experiment period, 24 pigs (six pigs per group) were randomly selected to collect fresh feces from the rectum. The feces samples (*n* = 48) were immediately frozen in liquid nitrogen until DNA extraction. Total bacterial genomic DNA was extracted using a stool DNA kit (Omega, Bio-Tek Inc., Norcross, GA, USA) according to the manufacturer’s protocol. The hypervariable region V3–V4 of the microbial 16S rRNA gene (accession numbers: SAMN26994143 to SAMN26994190) was amplified by PCR with indices and adaptors-linked universal primers (F: 5′-ACTCCTACGGGAGGCAGCAG-3′; R: 5′-GGACTACHVGGGTWT-CTAAT-3′). The PCR products were confirmed with 2% agarose gel electrophoresis, purified with AMPure XT beads (Beckman Coulter Genomics, Danvers, MA, USA), and then quantified by an Invitrogen Qubit 4.0 fluorometer (Invitrogen, Thermo Fisher Scientific, Waltham, MA, USA). The amplicon pools were prepared for sequencing, and the quantity of the amplicon library was assessed on the Agilent 2100 bioanalyzer (Agilent Technologies, Palo Alto, CA, USA) and the Library Quantification Kit for Illumina (Kapa Biosciences, Woburn, MA, USA). Amplicon libraries were sequenced on the Illumina MiSeq platform (Illumina, San Diego, CA, USA) for paired-end sequencing of 2 × 300 bp reads according to the manufacturer’s recommendations. The raw paired-end reads were truncated by removing the barcode and primer sequence. Quality filtering on the raw tags was performed to obtain high-quality clean tags using QIIME2 software [32]. The high-quality clean tags were clustered into operational taxonomic units (OTUs) with a similarity threshold of 0.97. Representative sequences were selected for each OUT, and the RDP classifier was used to further classify OTUs with representative sequences at an 0.80 confidence level [33]. Principal coordinate analysis (PCoA) plots were generated according to the unweighted UniFrac distance metrics [34]. The number of observed species and the indices of Chao 1 (species richness), as well as Shannon and Simpson (diversity), were calculated to estimate alpha diversity using QIIME 2 [32].

### 2.6. Statistical Analyses

Data of growth performances, the CBC test, antioxidant indicators, and fecal microbiota were analyzed using the GLIMMIX procedure of SAS 9.4 (SAS Institute Inc., Cary, NC, USA) with treatment, gender, time, and their interaction as fixed effects, and pen as a random effect according to the completely randomized design. Appropriate post-test comparisons for means were made for multiple groups using the Bonferroni Multiple Comparisons Test. A partial correlation analysis between the gut microbiota and antioxidant activity and growth performance indicators was carried out using R statistical software. The results are expressed as means ± standard error of the mean (SEM). Differences were considered significant at *p* < 0.05, while differences were considered to show a tendency at 0.05 ≤ *p* < 0.10.

## 3. Results

### 3.1. Growth Performance

No serious adverse events were observed during the whole experiment period. The effects of acidifiers on the growth performances of weaned pigs are presented in Table 1. The FBW in the acidifier groups showed an increased trend compared with that in the control group (0.05 < *p* < 0.10). The ADG of the A1 group was greater than that of the control group (*p* < 0.05). The ADFI of the A1 group was also greater than that of the control group (*p* < 0.05). However, no significant difference was found in the F:G between groups (*p* > 0.05).

### 3.2. Diarrhea and Survival Rate

The effects of acidifiers on the diarrhea and survival rate of weaned pigs are shown in Table 2. There was no significant difference in diarrhea rates between the acidifier treatment groups and the control group during the experimental period (*p* > 0.05). Nonetheless, the diarrhea rates of pigs in the A1, A2, and A3 groups were decreased by 7.5%, 2.9%, and 6.5% compared with the control group during the entire experimental period, respectively. Although there were also no significant differences in the survival rate of pigs between the A1, A3, and control groups (*p* > 0.05), there was a significant improvement (*p* = 0.02) on the survival rate of pigs between the A2 group and the control group in the Multiple Comparisons test. In summary, drinking water supplementation with acidifiers had a greater survival rate in the A2 group and numerically lower diarrhea rates than the control group at the end of the trial period.

### 3.3. CBC Test Indicators

The effects of acidifiers on the CBC test indicators of weaned pigs are shown in Table 3. On d 18, Neu in the A1 and A2 groups was less than that in the A3 group (*p* < 0.05); Neu%, HCT, and MCV in the A1 and A2 groups were less than those in the control group (*p* < 0.05); MCH, MCHC, and RDW-SD in the A1 and A2 groups were greater than those in the control group (*p* < 0.05); Mon % in the A1 group was greater than those in the other three groups (*p* < 0.05); EOS, RBC, MPV, and PDW in the A1 group were less than those in the control group (*p* < 0.05); RDW-CV in the A1 group was greater than those in control group (*p* < 0.05); LYM% in the A2 group was greater than that in the control group (*p* < 0.05). There was no significant difference in other blood routine indices among the four groups (*p* > 0.05).

Furthermore, on d 35, Mon of the A2 group was greater than that of the control group (*p* < 0.05); EOS in the A1 and A2 groups was greater than that in the control group (*p* < 0.05); EOS% in the A1 and A3 groups was greater than that in the control group (*p* < 0.05); PCT% in the A2 group was greater than that in the control group (*p* < 0.05); RBC and HCT in the A3 group were less than those in the control group (*p* < 0.05); MPV of the three acidifier groups was greater than that of the control group (*p* < 0.05). There were no significant differences in other blood routine indices among the four groups (*p* < 0.05).

### 3.4. Antioxidant Capacity Level

The effects of acidifiers on the level of antioxidant capacity of weaned pigs are shown in Table 4. The level of SOD in the A2 group was greater than that of the control group on d 18 (*p* < 0.05). The activities of CAT in the A1 and A2 groups were greater than the control group on d 18 (*p* < 0.05). The activities of GSH in the A1 and A3 groups were greater than the control on d 18 (*p* < 0.05). The level of T-AOC in the acidifier groups was greater than that of the control group on d 18 (*p* < 0.05), whereas the level of MDA in the acidifier supplementation groups was significantly lower than that in the control group on d 18. The activity of GSH in the acidifier groups was greater than that of the control group on d 35 (*p* < 0.05). The level of T-AOC in the A1 group was greater than that of the other three groups on d 35 (*p* < 0.05), whereas no significant difference was found in the activity of MDA between groups on d 35 (*p* > 0.05). Interestingly, the level of GSH-Px showed a tendency to increase in the A1 group when compared to control group on d 35 (0.05 < *p* < 0.10).

### 3.5. Microbiological Analysis of Gastrointestinal Contents

#### 3.5.1. Alpha Diversity Analysis

There were 56,867 to 79,927 valid reads obtained from each sample (Appendix A). Alpha diversity analysis including Observed species, Good’s coverage, Shannon, Simpson, and Chao1 indices are presented in Figure 1. The Shannon index of the A2 group was less than that of the A3 group on d 18 (*p* < 0.05). The observed OTUs (species) and Chao1 indices of the A3 group were less than those of the control group on d 35 (*p* < 0.05). The Simpson of the A1 group was less than that of the control group (*p* < 0.05), whereas the Good’s coverage index of the A1 and A3 groups was greater than that of the control group on d 35 (*p* < 0.05).

#### 3.5.2. Analysis of Group Differences of Intestinal Flora at the Phylum Level

The composition of intestinal flora at the phylum level is presented in Figure 2A and Appendix A. The most abundant phylum across all groups was *Firmicutes* at the phylum level, followed by *Bacteroidetes*, *Actinobacteria*, and *Proteobacteria*. The relative abundance of *Firmicutes* in intestinal flora on d 18 and d 35 were 75.86% and 84.12%, respectively. Next, we compared the differences of intestinal flora at the phylum level between treatments. The relative abundance of *Firmicutes* in the A1 and A2 groups was greater than that in the A3 group on d 18 (*p* < 0.05) (Figure 3A). The relative abundance of *Firmicutes* in the A1 group was greater than that in the control group on d 35 (*p* < 0.05) (Figure 3B). The relative abundance of *Proteobacteria* in the A2 group was less than that in the A3 group on d 35 (*p* < 0.05) (Figure 3H).

#### 3.5.3. Analysis of Group Differences of Intestinal Flora at the Genus Level

The composition of intestinal flora at the genus level in each group is presented in Figure 2B and Appendix A. The most abundant genus across all groups was *Lactobacillus*, accounting for 20.99% and 8.94% on d 18 and 35, respectively. The genus level analysis showed that the relative abundance of *Lactobacillus* in the A1 group was greater than that of the other three groups (*p* < 0.05); the relative abundance of *Lachnospiraceae_unclassified* in the A2 group was less than that of the control group or the A3 group on d 18 (*p* < 0.05) (Figure 4A). The relative abundance of *Lactobacillus* in the A1 group was greater than that of the control group and the A3 group on d 35 (*p* < 0.05); the relative abundance of *Streptococcus* in the A1 group was greater than that of the control group on d 35 (*p* < 0.05); the relative abundance of *Subdoligranulum* in the A3 group was greater than that of the control group on d 35 (*p* < 0.05); the relative abundance of *Solobacterium* in the A3 group was greater than that of other three groups on d 35 (*p* < 0.05); the relative abundance of *Ruminococcaceae_UCG-005* in the A2 group was greater than that of the control group and the A3 group on d 35 (*p* < 0.05) (Figure 4B).

### 3.6. Partial Correlation Analyses between the Differential Microbial Species and Measured Parameters

The results of partial correlation analyses are presented in Figure 5. The results showed that *Streptococcus* had a significant positive correlation with serum CAT level (R = 0.998, *p* < 0.05). *Subdoligranulum* had a significant positive correlation with ADG and ADFI (R = 0.92, *p* < 0.05), while having a negative correlation with diarrhea rate (R = −0.88, *p* < 0.05). *Ruminococcaceae_UCG-005* was significantly positively correlated with serum CAT, GSH, GSH-Px, and T-AOC level (R = 0.99–1.00, *p* < 0.05), while negatively correlated with MDA concentration (R = −1.00, *p* < 0.05). There were no significant associations found in other microbiota (*p* > 0.05).

## 4. Discussion

The digestive system of weaned pigs is immature, and the secretion of gastric acid and digestive enzymes is insufficient, resulting in the inability to activate digestive enzymes such as pepsin, making them unable to effectively digest nutrients such as protein and starch in the diet [35]. Therefore, improving the digestive capacity and environment of pigs plays a vital role in promoting the growth performance of pigs. Supplementation with a microencapsulated blend of organic acids improved the feed intake, average daily gain, and weight gain rate of weaned pigs [36]. Yang et al. [37] also found that adding the mixture of essential oils and organic acids to the diet improved the final weight and daily gain of weaned pigs. Several studies have reported the beneficial effects of compound organic acids (formic acid, acetic acid, propionic acid, and lactic acid) in swine feed [38]. However, due to the type and dose of dietary organic acids, coating or mixing mode, dietary composition, and other factors, the response of dietary organic acids to weaned pigs is different. Compared with supplementation in feed, drinking water supplementation with acidifiers is more convenient and makes it easier to control the dosage in pig farms. In addition, the acidifiers can disinfect drinking water and inhibit pathogenic bacteria. Meanwhile, few studies have investigated the effects of different organic acid combinations supplied via the water in the post-weaning period of piglets. Here, we present, for the first time, the synergistic effect of an organic acid formula (containing formic, acetic, and propionic acids) with or without lactic acid in pigs. In our present study, the ADG and ADFI of the A1 group was greater than that of the control group, while there was no significant difference between the A2 and A3 groups and the control group. The organic acids combination of the A1 group contains lactic acid compared with the A2 and A3 groups. A study showed that the addition of compound organic acids containing formic acid and lactic acid significantly reduced the *Salmonella* seroprevalence compared with the addition of formic acid alone in the feed of fattening pigs [39]. It has been previously reported that lactic acid reduced the pH value of the gastric juices and inhibited the reproduction of enterotoxin *Escherichia coli*, which is more effective than other organic acids in improving the growth performance of pigs [40], which is basically consistent with our results. The effect of compound organic acid as an alternative to antibiotics is better than a single organic acid in weaned pigs [41]. This indicates that the synergistic blend of formic acid, acetic acid, propionic acid, and lactic acid could improve intestinal health, appetite, and feed intake, so as to improve the utilization rate of feed and make the weight gain of weaned pigs significant.

Moreover, piglets are subjected to nutritional, environmental, and psychological stress during the weaning period, which leads to post-weaning diarrhea syndrome (PWDS) [42]. When the degree of diarrhea is mild, it causes malnutrition and affects the growth and development of piglets [43]. When it is serious, it causes dehydration and even death in piglets [44]. A previous study found that dietary supplementation with an acidifier inhibited the reproduction of ETEC F4 and ETEC K99, and reduced the severity and duration of diarrhea in weaned pigs [45]. In the present study, the diarrhea rate of pigs in the experimental groups were numerically less than that in the control group, but the difference was not significant. This might be due to the addition of the acidifier to the drinking water having less of an effect on piglet diarrhea than adding the acidifier to feed. On the other hand, the weaned pigs were raised in a modern nursery and had an acclimation period in the nursery before the experiment, which might also be one of the reasons why the diarrhea rate was not significantly different between the acidifier group and the control group. Interestingly, there was significant improvement on the survival rate of pigs between the A2 group and the control group in the Multiple Comparisons, and the survival rate of other acidifier groups was also numerically greater than that of the control group. Most pigs that died were small and weak during the whole experimental period. The reestablishment of social hierarchy after mixing causes weak pigs to be at a disadvantage in competing for feed, while social competition has no effect on drinking water behavior [46,47]. It suggests that adding acidifiers to the drinking water can improve the survival rate of weak piglets.

When animals are healthy, there is a dynamic balance between the production of free radicals and the ability of the antioxidant system to scavenge free radicals in the body [48]. Post-weaning piglets produced too many oxides and free radicals, resulting in an imbalance of the redox potential and oxidative stress damage, which affects the immune response and growth performance of piglets [49,50]. Therefore, it is very important to improve the antioxidant stress ability of weaned pigs by eliminating free radicals and regulating the balance and stability of redox potential through nutritional intervention. Oxidative stress was determined by detecting the activities of glutathione peroxidase (GPX) and superoxide dismutase (SOD) in piglets [51]. In addition, genes related to oxidative stress, including catalase (CAT), lactate dehydrogenase (LDHA), glutathione peroxidase 2 (GPX2), and superoxide dismutase 3 (SOD3), often changed during the weaning transition period of piglets [52]. Organic acid-based feed additives improved the levels of GSH and ferric-reducing ability potential (FRAP) in the ileum, and had a significant antioxidant effect [53]. Furthermore, supplementing the water plus organic acid blends to the basal diet of piglets had significantly increased serum T-AOC activities [54]. As an immunoassay technique, ELISA has been widely used to detect and quantify proteins, antibodies, or hormones [55]. An ELISA-based competition assay has the advantages of high sensitivity, simple operation, and affordability [56], exhibiting greater effectiveness than the HPLC-FLD method [57]. The FRAP assay (Ferric Reducing Ability of Plasma), a simple method to determine the total antioxidant capacity, has been applied to detect the T-AOC of animal serum [58]. The FRAP assay has the advantages of being inexpensive, has a simple reagent preparation, and high reproducibility [59]. In the present experiment, the concentrations of MDA, SOD, GSH, GSH-Px, and CAT were measured by ELISA-based competition assays. The results showed that adding acidifiers to the drinking water significantly increased the T-AOC level of weaned pigs, which was consistent with the above research results. This indicated that adding acidifiers to the drinking water effectively enhanced the antioxidant capacity and reduced oxidative stress in weaning pigs.

The stability of intestinal flora has long been known to play a vital role to maintain the metabolic health of the host [60]. If the microflora was disordered, it reduced the immunity of the animals and caused a variety of diseases [61]. The structure of the intestinal flora of adult animals is generally considered to be stable. Meanwhile, the bacteria in piglets mainly comes from the mother from newborn to pre-weaning, and the intestinal micro ecosystem remains relatively balanced [62]. However, the food of weaned pigs is changed from liquid breast milk to solid feed, coupled with the changes of environmental factors such as humidity, temperature, and population transformation, the harmful flora in the intestine of piglets increases, which destroys the balance of intestinal flora and affects the growth and development of the host [63]. Therefore, it is of great significance to find feed additives that improve the intestinal microflora of weaned pigs and maintain the intestinal health of animals. Acidifiers reduced the pH of the gastrointestinal tract and created an acidic environment, which might be helpful in improving the gastrointestinal environment of piglets [64]. Observed species, Chao1, Shannon, and Simpson indices are indicators reflecting alpha diversity [65]. In this study, the 16S rRNA gene sequencing of the intestinal contents of piglets showed that the Observed species and Chao1 index of the A3 group were less than those of the control group on d 35. The Simpson index of the A1 group was less than that of the control group on d 35. The above results showed that the number of microbial species in the feces of weaned pigs decreased significantly after adding acidifiers to the drinking water for 35 d. This suggests that acidifiers might regulate the pH value of the gastrointestinal tract and inhibit the growth of harmful bacteria in the gastrointestinal tract of weaned pigs, so as to improve the structure of the gastrointestinal flora of piglets.

In addition to its efforts on microbial richness and diversity, this study had indicated that supplementing the drinking water with acidifiers also exerted an effect on the abundance of intestinal flora in piglets. In this experiment, *Firmicutes* and *Bacteroidetes* were the dominant bacteria in the intestinal microorganisms of piglets, and their relative abundance was high, which was basically consistent with previous experimental results [66,67]. The relative abundance of *Firmicutes* in the A1 group was the greatest on d 18 and 35, and also greater than that of the control group on d 35, which indicated that acidifier 1 could improve the relative abundance of *Firmicutes* in piglets. *Firmicutes* was reported to be related to the weight gain of weaned pigs [68]; however, there was no significant correlation between *Firmicutes* and the daily weight gain of weaned pigs in this study. It might be that the increase of *Firmicutes* abundance improved antioxidant capacity and indirectly improved growth performance, which might be the reason for the significant increase of ADG in the A1 group. In addition, Li et al. [69] found that when the compound acidifier was composed of formic acid, acetic acid, propionic acid, and medium chain fatty acids were added to the diet of growing pigs, the content of *Bacteroides* in the cecum of growing pigs decreased by 8.8%. In this study, the relative abundance of *Bacteroidetes* in the three experimental groups were less than that of the control group on d 35, which is similar to the results of the above research.

All bacteria belonging to *Proteobacteria* are Gram-negative bacteria, including *Salmonella*, *Escherichia coli*, and other pathogenic bacteria [70], which seriously affect the health of animals [71]. In this study, the relative abundance of *Proteobacteria* in the A2 group was the lowest among the four groups on d 35, indicating that acidifier 2 had the trend of reducing the relative abundance of *Proteobacteria* in pigs. *Lactobacillus* belongs to Gram-positive bacteria, which regulates the balance of intestinal flora of animals, enhances immunity, and improves feed digestibility [72]. Studies have shown that acidifiers also increased the abundance of *Lactobacillus* in the feces of weaned pigs [73]. The relative abundance of *Lactobacillus* in the A1 group was greater than that of the other three groups on d 18; the relative abundance of *Lactobacillus* in the A1 group was greater than that of the control group and A3 group on d 35. This might be because acidifier 1 contains lactic acid, which increased the relative abundance of *Lactobacillus* in the intestine of piglets, allowing *Lactobacillus* to became the dominant flora, thus improving the micro ecological balance of the intestinal environment. Therefore, adding acidifier 1 to the drinking water improved the intestinal flora structure of weaned pigs, so as to regulated the balance of gastrointestinal flora.

Partial correlation is used to calculate the strength of the relationship between two variables while accounting for the effects of one or more other variables. The partial correlation analyses showed that the relative abundance of *Subdoligranulum* was positively correlated with ADG and ADFI, while negatively correlated with diarrhea rate. The relative abundance of *Subdoligranulum* was lower in the colon of Ningxiang pigs (a fatty-type Chinese Indigenous pig breed) compared to that of Large White pigs [74]. *Subdoligranulum* has a positive effect on fecal microbiota transplantation in the treatment of necrotizing enterocolitis by affecting the production of butyrate [75]. In the present study, the relative abundance of *Subdoligranulum* in the A3 group was greater than that of the control group. The relative abundances of *Subdoligranulum* in the A1 and A2 groups were also numerically greater than that of the control group on d 35. We speculated that the addition of the acidifier to drinking water improves the abundance of *Subdoligranulum*, which contributes to improve intestinal health, increase feed intake, and daily weight gain, while decreasing diarrhea rate. The gut is a target site of reactive oxygen species (ROS) accumulation and consequent oxidative stress [76]. The partial correlation analysis showed that *Ruminococcaceae_UCG-005* was positively correlated with serum CAT, GSH, GSH-Px, and T-AOC levels, whereas it was negatively correlated with MDA concentration. Feeding gallic acid increased catalase and the total antioxidant capacity levels, while decreasing malondialdehyde concentrations and increasing the relative abundances of *Ruminococcaceae_UCG-005* in preweaning calves [77]. Rodents drinking silicon-containing water (BT) with antioxidant activity increased plasma H_2_O_2_ scavenging activity and glutathione peroxidase activity, while significantly increasing the abundance of *Ruminococcaceae_UCG-005*, which is basically consistent with the present study [78]. Our study showed that the relative abundance of *Ruminococcaceae_UCG-005* in the A2 group was significantly greater than that of the control group, and the A1 group was also numerically greater than the control group on d 35. It indicates that *Ruminococcaceae_UCG-005* may play an antioxidant role when piglets are subjected to weaning stress. Altogether, the disruption of gut microbial composition could be the major underlying factor inducing the decline in the antioxidant capacity of weaning-challenged piglets. These results contribute to the new understanding of the acidifier-enhanced antioxidant capacity, at least in part, due to alterations in the gut microbiota in weaned pigs.

## 5. Conclusions

In conclusion, the current findings suggested that adding acidifier 1 (19% formic acid +19% acetic acid +3.5% propionic acid +15% lactic acid) to the drinking water of weaned piglets significantly improved the ADG and ADFI of weaned pigs. In addition, we also found that adding acidifiers to the drinking water significantly improved the total antioxidant capacity of serum in weaned pigs. The addition of acidifier 1 to the drinking water increased the relative abundance of *Firmicutes* and *Lactobacillus* in the intestines of pigs, and improved the community structure of intestinal flora of pigs. These results also demonstrated the potential of acidifiers as an alternative to antibiotics in promoting the growth of weaned pigs, improving the ability of antioxidant capacity, and improving the intestinal microflora.

## Figures and Tables

**Figure 1 antioxidants-11-00809-f001:**
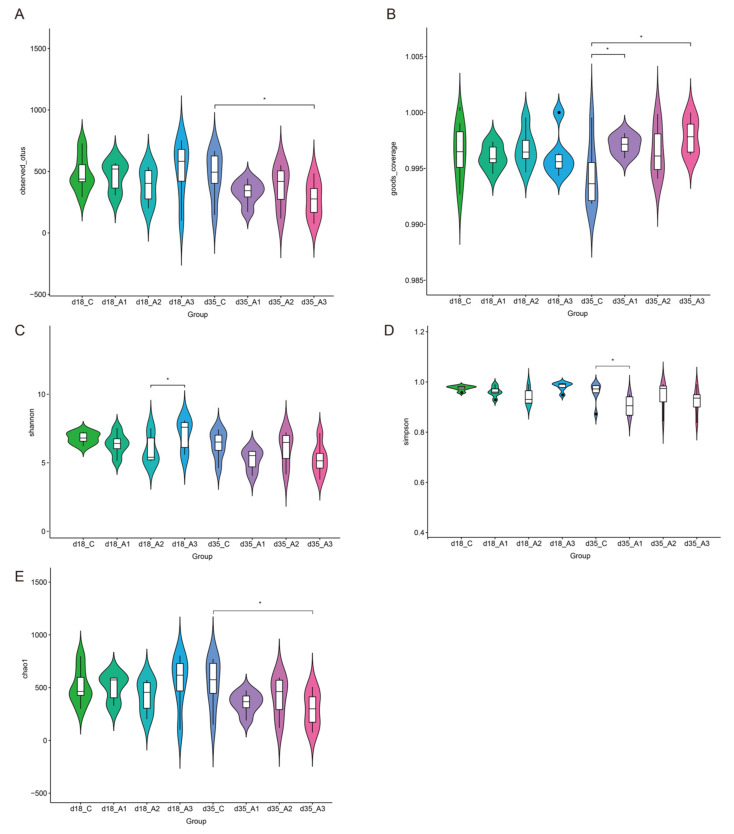
Comparison of the Alpha diversity index in the four groups on d 18 and d 35. Observed OTUs (observed species) (**A**). Good’s coverage (**B**). Simpson indices (**C**). Shannon indices (**D**). Chao1 indices (**E**). Labeled with * indicates a significant difference, *p* < 0.05; Mean values are based on six pigs (six pigs per group).

**Figure 2 antioxidants-11-00809-f002:**
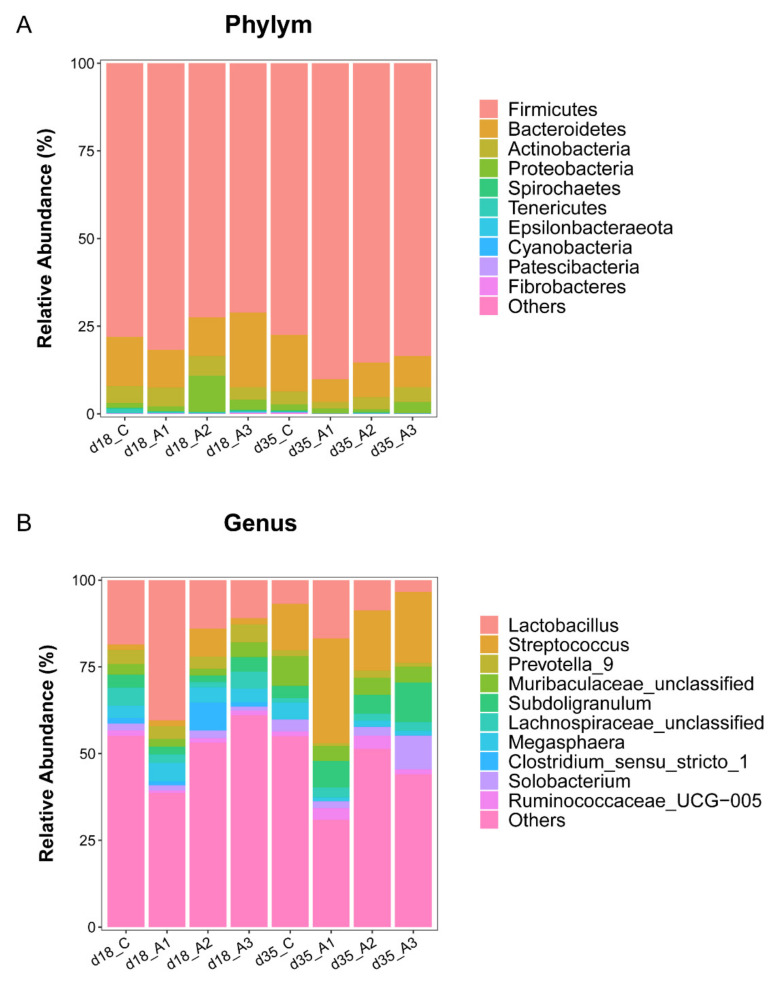
Gut microbiota composition of experimental group pigs and control group pigs: The relative abundance of the top 10 intestinal flora at the phylum level (**A**); the relative abundance of the top 10 intestinal flora at the genus level (**B**).

**Figure 3 antioxidants-11-00809-f003:**
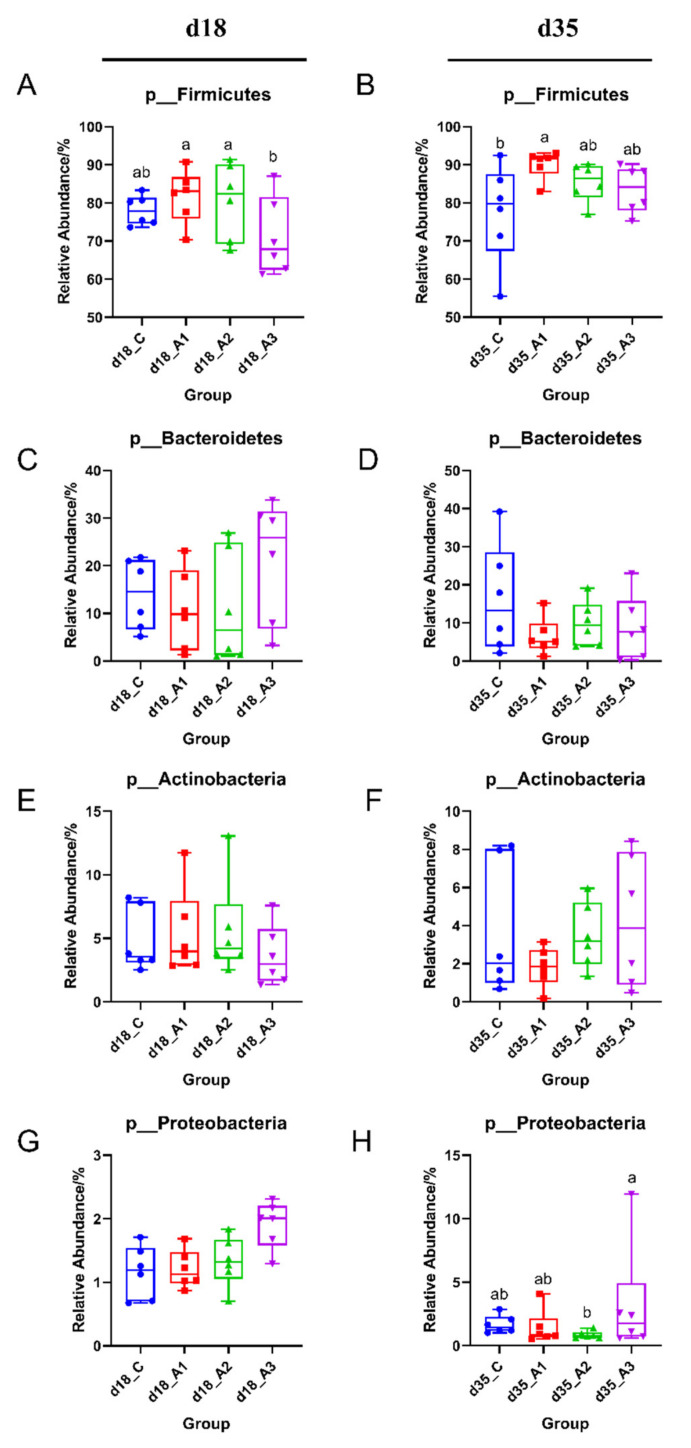
Comparison of dominant intestinal flora at the phylum level. Relative abundances of *Firmicutes* (**A**), *Bacteroidetes* (**C**), *Actinobacteria* (**E**), and *Proteobacteria* (**G**) among the four groups on d 18. Relative abundances of *Firmicutes* (**B**), *Bacteroidetes* (**D**), *Actinobacteria* (**F**), and *Proteobacteria* (**H**) among the four groups on d 35. Labeled means with different superscript letters are significantly different, *p* < 0.05. Mean values are based on six pigs (six pigs per group).

**Figure 4 antioxidants-11-00809-f004:**
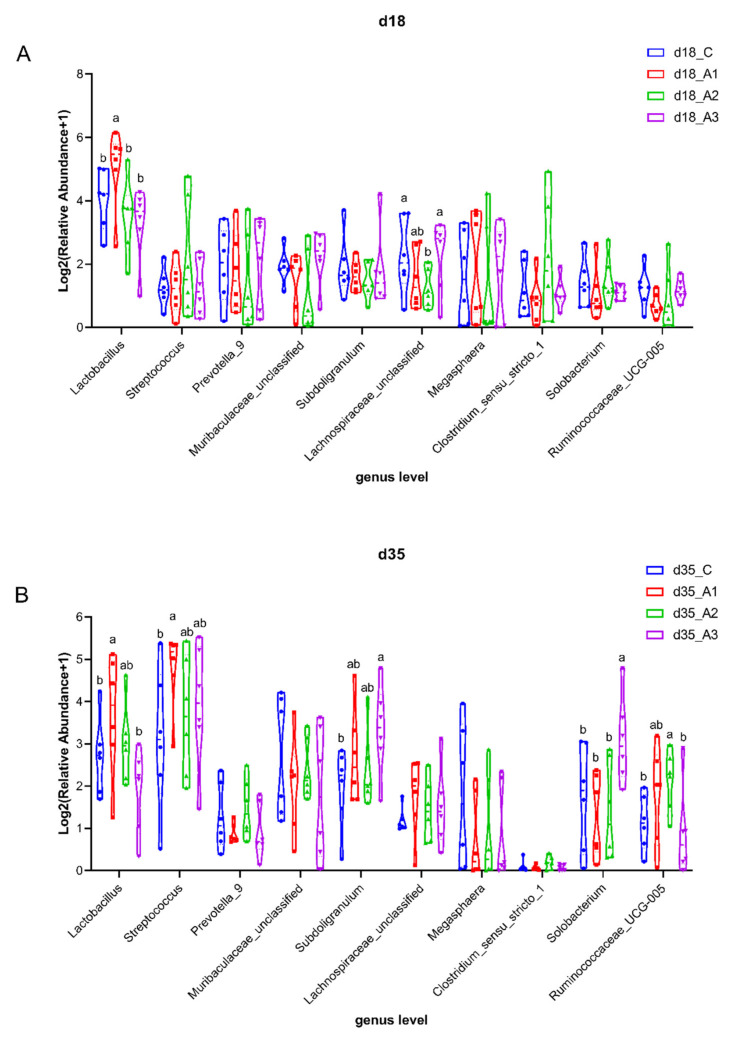
Comparison of dominant intestinal flora at the genus level. Relative abundances of dominant intestinal flora at the genus level among the four groups on d18 (**A**). Relative abundances of dominant intestinal flora at the genus level among the four groups on d18 (**B**). Labeled means with different superscript letters are significantly different, *p* < 0.05. Mean values are based on six pigs (six pigs per group).

**Figure 5 antioxidants-11-00809-f005:**
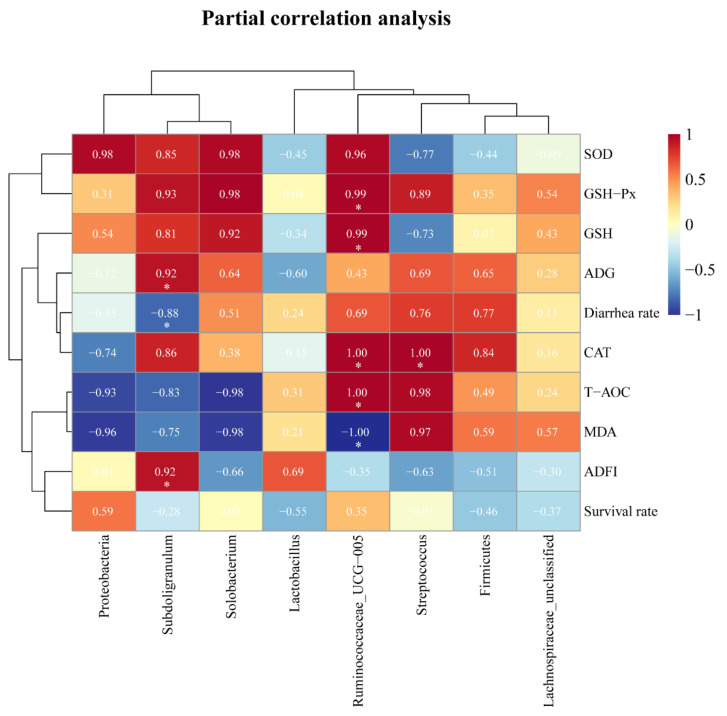
The partial correlation analyses between the differential microbial species and measured parameters. ADG, average daily gain; ADFI, Average daily feed intake; SOD, total superoxide dismutase; CAT, catalase; GSH, glutathione; GSH-Px, glutathione peroxidase; T-AOC, total antioxidant capacity; MDA, malondialdehyde; The red represents positive correlation, blue represents negative correlation, and the depth of the color represents the degree of correlation, respectively (Labeled with * indicate significantly different, *p* < 0.05).

**Table 1 antioxidants-11-00809-t001:** Effects of acidifiers on the growth performances of weaned pigs ^1^.

Items	Treatment ^2^				SEM ^3^	*p*-Value
	Control	A1	A2	A3		
IBW, kg	9.20 ^c^	10.41 ^a^	10.02 ^b^	9.05 ^c^	0.14	0.002
FBW, kg	24.81 ^b^	27.16 ^a^	26.01 ^a,b^	25.38 ^a,b^	0.42	0.091
ADG, g	445.29 ^b^	512.25 ^a^	478.81 ^a,b^	462.26 ^b^	12.85	0.040
ADFI, g	645.40 ^b^	743.80 ^a^	696.20 ^a,b^	653.50 ^b^	11.35	0.024
F:G	1.45	1.45	1.46	1.41	0.009	0.130

IBW, Initial body weight; FBW, Final body weight; ADG, average daily gain; ADFI, average daily feed intake; G:F, gain-to-feed ratio. ^a,b,c^ Different superscript within a row indicate a significant difference (*p* < 0.05). ^1^ Covariance analysis was used to correct the effect of different initial weights on average daily gain. The initial weight was corrected to 9.7026 kg. ^2^ A1, acidifiers A1 group, continuous supply of acidified drinking water with 19% formic acid +19% acetic acid +3.5% propionic acid +15% lactic acid; A2, acidifiers A2 group, continuous supply of acidified drinking water with 22% formic acid +16% acetic acid +4% propionic acid; A3, acidifiers A3 group, continuous supply of acidified drinking water with 22% formic acid +16% acetic acid +4% propionic acid +0.5% stevioside; control, continuous supply of non-treated water. ^3^ SEM means standard error of the means (*n* = 100).

**Table 2 antioxidants-11-00809-t002:** Effects of acidifiers on the diarrhea and survival rate of weaned pigs.

Items	Treatment ^1^				SEM ^2^	*p*-Value
	Control	A1	A2	A3		
Diarrhea rate, %						
d 1 to 18	3.11	2.72	2.95	2.78	0.30	0.802
d 19 to 35	3.00	2.94	3.00	2.94	0.23	0.995
d 1 to 35	3.06	2.83	2.97	2.86	0.25	0.905
Survival rate, %						
d 1 to 35	93.00 ^b^	96.00 ^a,b^	99.00 ^a^	95.00 ^a,b^	1.56	0.107

^a,b^ Different superscripts within a row indicate a significant difference (*p* < 0.05). ^1^ A1, acidifiers A1 group, continuous supply of acidified drinking water with 19% formic acid +19% acetic acid +3.5% propionic acid +15% lactic acid; A2, acidifiers A2 group, continuous supply of acidified drinking water with 22% formic acid +16% acetic acid +4% propionic acid; A3, acidifiers A3 group, continuous supply of acidified drinking water with 22% formic acid +16% acetic acid +4% propionic acid +0.5% stevioside; control, continuous supply of non-treated water. ^2^ SEM means standard error of the means (*n* = 100).

**Table 3 antioxidants-11-00809-t003:** Effects of acidifiers on the complete blood count (CBC) test of weaned pigs.

Items ^1^	Treatment ^2^				SEM ^3^	*p*-Value
	Control	A1	A2	A3		
d18						
WBC, 10^9^/L	21.17	18.56	17.92	20.50	2.09	0.653
Neu, 10^9^/L	6.59 ^a,b^	2.74 ^b^	2.96 ^b^	8.65 ^a^	1.39	0.007
LYM, 10^9^/L	12.34	12.50	12.98	10.18	1.41	0.507
Mon, 10^9^/L	1.79 ^a,b^	3.12 ^a^	1.84 ^a,b^	1.26 ^b^	0.55	0.109
Eos, 10^9^/L	0.37 ^a^	0.15 ^b^	0.13 ^b^	0.31 ^a,b^	0.07	0.057
Bas, 10^9^/L	0.09	0.06	0.07	0.10	0.02	0.407
Neu, %	31.52 ^a^	13.93 ^b^	14.08 ^b^	37.08 ^a^	2.48	<0.001
LYM, %	57.66 ^b,c^	68.46 ^a,b^	73.65 ^a^	55.00 ^c^	4.42	0.010
Mon, %	8.69 ^c^	15.48 ^a^	11.46 ^b^	6.01 ^c^	1.18	<0.001
Eos, %	1.77 ^a^	0.89 ^b,c^	0.56 ^c^	1.50 ^a,b^	0.27	0.008
Bas, %	0.36 ^a,b^	0.26 ^b^	0.26 ^b^	0.41 ^a^	0.05	0.042
RBC, 10^9^/L	5.30 ^a^	4.35 ^b^	4.53 ^b^	5.63 ^a^	0.25	<0.001
HGB, g/L	100.56	103.30	103.01	104.92	5.45	0.366
HCT, %	30.57 ^a^	19.80 ^b^	21.58 ^b^	31.89 ^a^	1.40	<0.001
MCV, fL	57.66 ^a^	45.50 ^b^	46.79 ^b^	56.18 ^a^	0.85	0.000
MCH, pg	18.96 ^b^	22.20 ^a^	21.83 ^a^	18.53 ^b^	0.58	<0.001
MCHC, g/L	329.06 ^c^	477.67 ^a^	462.67 ^b^	330.13 ^c^	12.99	0.000
RDW-CV, %	19.41 ^c^	36.23 ^a^	33.28 ^b^	19.23 ^c^	0.59	0.000
RDW-SD, fL	38.99 ^b^	57.04 ^a^	54.21 ^a^	37.59 ^b^	1.01	0.000
PLT, 10^9^/L	247.19	371.25	384.98	212.00	65.61	0.003
MPV, fL	9.16 ^a^	7.99 ^b^	8.28 ^b^	9.25 ^a^	0.29	0.001
PDW, %	15.53 ^a^	14.95 ^c^	15.04 ^b,c^	15.43 ^a,b^	0.16	0.023
PCT, %	0.23 ^a,b^	0.18 ^b^	0.27 ^a^	0.20 ^a,b^	0.04	0.106
d35						
WBC, 10^9^/L	26.02 ^a,b^	29.77 ^a^	31.40 ^a^	22.60 ^b^	2.09	0.017
Neu, 10^9^/L	9.39	12.46	11.76	8.85	1.39	0.194
LYM, 10^9^/L	12.93	11.91	13.77	9.90	1.41	0.246
Mon, 10^9^/L	2.98 ^b^	4.40 ^a,b^	3.76 ^a^	3.04 ^b^	0.55	0.034
Eos, 10^9^/L	0.58 ^b^	0.87 ^a^	0.87 ^a^	0.73 ^a,b^	0.07	0.019
Bas, 10^9^/L	0.15 ^a^	0.13 ^a,b^	0.15 ^a^	0.09 ^b^	0.02	0.061
Neu, %	34.43	40.69	37.79	37.59	1.36	0.761
LYM, %	51.39	41.92	44.96	44.99	4.42	0.489
Mon, %	11.49	14.03	12.08	13.74	1.18	0.360
Eos, %	2.19 ^b^	2.96 ^a^	2.76 ^a,b^	3.34 ^a^	0.27	0.026
Bas, %	0.51 ^a^	0.41 ^a,b^	0.45 ^a,b^	0.34 ^b^	0.02	0.078
RBC, 10^9^/L	5.78 ^a^	6.03 ^a^	5.91 ^a^	5.10 ^b^	0.10	0.006
HGB, g/L	105.31 ^a^	104.31 ^a^	104.75 ^a^	92.81 ^b^	5.45	0.098
HCT, %	34.68 ^a^	36.34 ^a^	34.51 ^a^	29.23 ^b^	1.40	0.003
MCV, fL	59.99	60.33	58.54	59.71	0.85	0.473
MCH, pg	18.24	17.36	17.76	18.29	0.58	0.626
MCHC, g/L	304.75	287.81	303.94	307.12	12.99	0.709
RDW-CV, %	20.13	20.62	20.86	20.07	0.59	0.740
RDW-SD, fL	42.36	43.53	43.13	41.84	1.01	0.646
PLT, 10^9^/L	227.00	305.00	346.25	256.62	65.61	0.588
MPV, fL	9.03 ^b^	10.07 ^a^	10.68 ^a^	9.99 ^a^	0.29	0.001
PDW, %	16.28	16.13	15.84	16.19	0.16	0.229
PCT, %	0.22 ^b^	0.32 ^a,b^	0.37 ^a^	0.28 ^a,b^	0.04	0.048

^a,b,c^ Different superscripts within a row indicate a significant difference (*p* < 0.05). ^1^ WBC, white blood cell; Neu, neutrophil count; LYM, lymphocyte; Mon, monocyte; Eos, eosinophilic; Bas, basophil; RBC, red blood cell; HGB, hemoglobin; HCT, hematocrit; MCV, mean corpuscular volume; MCH, mean corpuscular hemoglobin; MCHC, mean corpuscular hemoglobin concentration; RDW-CV, red cell distribution width-coefficient of variation; RDW-SD, red cell distribution width-standard deviation; PLT, platelets; MPV, mean platelet volume; PDW, platelet distribution width; PCT, thrombocytocrit. ^2^ A1, acidifiers A1 group, continuous supply of acidified drinking water with 19% formic acid +19% acetic acid +3.5% propionic acid +15% lactic acid; A2, acidifiers A2 group, continuous supply of acidified drinking water with 22% formic acid +16% acetic acid +4% propionic acid; A3, acidifiers A3 group, continuous supply of acidified drinking water with 22% formic acid +16% acetic acid +4% propionic acid +0.5% stevioside; control, continuous supply of non-treated water. ^3^ SEM means standard error of the means (*n* = 16).

**Table 4 antioxidants-11-00809-t004:** Effects of acidifiers on antioxidant capacity in pigs.

Items ^1^	Treatment ^2^				SEM ^3^	*p*-Value
	Control	A1	A2	A3		
d18						
SOD, U/mL	141.76 ^b^	123.87 ^c^	162.43 ^a^	140.32 ^b^	3.51	<0.001
CAT, U/mL	6.28 ^b^	7.70 ^a^	8.25 ^a^	5.37 ^b^	0.37	< 0.001
GSH, μg/mL	162.03 ^b^	218.24 ^a^	150.95 ^b^	215.37 ^a^	10.14	<0.001
GSH-Px, U/mL	656.06	653.43	547.75	689.38	50.77	0.227
T-AOC, U/mL	4.08 ^c^	5.12 ^b^	5.74 ^a^	5.83 ^a^	0.19	<0.001
MDA, nmol/mL	4.69 ^a^	2.67 ^b^	2.75 ^b^	2.70 ^b^	0.19	<0.001
d35						
SOD, U/mL	118.29 ^a,b^	127.58 ^a^	115.74 ^b^	123.22 ^a,b^	3.51	0.087
CAT, U/mL	13.89 ^b^	14.80 ^a,b^	14.64 ^a,b^	14.97 ^a^	0.37	0.188
GSH, μg/mL	195.46 ^c^	236.22 ^b^	281.37 ^a^	227.00 ^b^	10.14	<0.001
GSH-Px, U/mL	436.38 ^b^	583.96 ^a^	432.92 ^b^	551.07 ^a,b^	50.77	0.076
T-AOC, U/mL	4.23 ^b^	5.44 ^a^	4.60 ^b^	4.41 ^b^	0.19	<0.001
MDA, nmol/mL	2.61	2.27	2.45	2.58	0.19	0.563

^a,b,c^ Different superscripts within a row indicate a significant difference (*p* < 0.05). ^1^ SOD, superoxide dismutase; CAT, catalase; GSH, glutathione; GSH–Px, glutathione peroxidase; T-AOC, total oxidative capacity; MDA, malonaldehyde. ^2^ A1, acidifiers A1 group, continuous supply of acidified drinking water with 19% formic acid +19% acetic acid +3.5% propionic acid +15% lactic acid; A2, acidifiers A2 group, continuous supply of acidified drinking water with 22% formic acid +16% acetic acid +4% propionic acid; A3, acidifiers A3 group, continuous supply of acidified drinking water with 22% formic acid +16% acetic acid +4% propionic acid +0.5% stevioside; control, continuous supply of non-treated water. ^3^ SEM means standard error of the means (*n* = 16).

## Data Availability

Data are contained within the article.

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
