# Peer review of "Drinking Water Supplemented with Acidifiers Improves the Growth Performance of Weaned Pigs and Potentially Regulates Antioxidant Capacity, Immunity, and Gastrointestinal Microbiota Diversity"

_antioxidants, 2022, doi:10.3390/antiox11050809_

Round 1

Reviewer 1 Report

Dear authors,

This manuscript (antioxidants-1664915) devoted to the evaluation of the “effects of acidified drinking water on the growth performance, immunity parameters, oxidative stress and intestinal microflora of weaned piglets” The authors are trying to compare acidifiers and to find out which type of acidifiers is “more suitable to be added to the drinking water of weaned piglets as an antibiotic substitute”.

The manuscript analyze the literature works in detail and at high level of discussion. I do not doubt the technical quality of the work and feel that there is a sufficient impact on a broader readership to justify publication in the "Antioxidants". This topic is in frame of the journal scope, the subject matter is treated in depth. The authors found that an adding of some acidifiers to the drinking water of weaned piglets significantly improve: 1) the ADG and ADFI of weaned piglets; 2) the total antioxidant capacity of serum in weaned piglets; 3) the relative abundance of Firmicutes and Lactobacillus in the intestine of piglets; the community structure of intestinal flora of piglets. The results (obtained by the authors) demonstrated “the potential of acidifiers as alternative to antibiotics on promoting the growth of weaned piglets, improving the ability of antioxidant stress and improving the intestinal microflora”.

Thus, the present manuscript is important and actual.

There are some comments:

  1. Lines 129-134. It is important to add details (such as the method of determination, wavelength for spectrophotometry or other important parameters, etc.) of the oxidative stress indicators measurements (lines 129-134), i.e. “malondialdehyde (MDA), total superoxide dismutase (SOD), glutathione (GSH), glutathione peroxidase (GSH-Px), catalase (CAT) and total antioxidant capacity (T-AOC)”. It is not enough to mention “….were determined using ELISA Kits (Shanghai Langdun Biotechnology Co., Ltd., Shanghai, China) ………..”. For example, I cannot found the details of the malondialdehyde (MDA) determination using different combination of the words “elisa kits manufacturers in China”, but only the following: https://www.made-in-china.com/productdirectory.do?word=Elisa+Kit+for+MDA&file=&searchType=0&subaction=hunt&style=b&mode=and&code=0&comProvince=nolimit&order=0&isOpenCorrection=1&org=top without details.
  2. On the other hand, there are a lot of methods of determination of the oxidative stress indicators. For example, there are at least three usual methods of the malondialdehyde (MDA) determination: traditional spectrophotometric determination of the malondialdehyde-thiobarbituric acid (MDA-TBA) complex (method A) and the two HPLC separation methods of the MDA-TBA (method B) and MDA-dinitrophenylhydrazine (MDA-DNPH) adduct (method C). The results of such measurements are more available in the literature as the measurements with “ELISA Kits”. It will be useful, if the authors will explain (in the part “4. Discussion”) the advantages of the of the oxidative stress indicators measurements with “ELISA Kits” in comparison to the usual methods of determination of these indicators, i.e. malondialdehyde (MDA) and others.
  3. Lines 369-370. Please, correct the sentence “This is basically consistent with the results of this study, indicating that acidifiers have a certain effect on reducing diarrhea in weaned piglets.” (Lines 369-370 in the part “4. Discussion”), because there are no pronounced effects on reducing diarrhea in weaned piglets (see Table 2 on line 203). In addition, the authors wrote “There was no significant difference in diarrhea rate between the four groups during the experimental period (p > 0.05).” (Lines 195-196).
  4. It will be useful, if the authors will explain (in the parts “3.4. Oxidative Stress Level” or “4. Discussion”) the data concerning the oxidative stress indicators (presented in the Table 4, lines 253-260). For example, there are extremely low and high GSH data of the A2 group (at day 18 and 35, respectively) as compared to the groups A1 and A3 (Table 4). In contrast, there are very high and relatively low T-AOC data of the A2 and A3 groups (at day 18 and 35, respectively) as compared to the groups A1 (Table 4), etc.
  5. There are the interesting data presented at the Figure 5 (Lines 331-335), but there are insufficient explanations of these correlations (between the differential microbial species and measured parameters). It is especially important because of the controversial correlations: strong (weak) or positive (negative) in the case of the similar oxidative stress indicators and the differential microbial species.
  6. Moderate editing of English language and style required. For example, lines 98-100. “There were 4 replicates (pen) in each group, and 25 piglets per pen. The concentration of acidifier is acidifier: water = 1:500, used for 8 h every day, free to drink”.

Please, consider my editing: “There were 4 replicates (pen) in each group, and 25 piglets per pen; used for 8 h every day; free to drink. The ratio of acidifier: water = 1:500 was used in this study”.

Author Response

Dear Reviewer 1,

Thank you very much for your constructive comments! We are appreciated that the Editor-in-Chief and you gave us an opportunity to revise our manuscript (antioxidants-1664915).

In the revised manuscript, we have meticulously addressed all the comments raised by the reviewers and associate editor. Our response to the comments was listed point-by-point. All the significant changes in the revised manuscript have been highlighted in red.

Point 1: Lines 129-134. It is important to add details (such as the method of determination, wavelength for spectrophotometry or other important parameters, etc.) of the oxidative stress indicators measurements (lines 129-134), i.e. “malondialdehyde (MDA), total superoxide dismutase (SOD), glutathione (GSH), glutathione peroxidase (GSH-Px), catalase (CAT) and total antioxidant capacity (T-AOC)”. It is not enough to mention “….were determined using ELISA Kits (Shanghai Langdun Biotechnology Co., Ltd., Shanghai, China) ………..”. For example, I cannot found the details of the malondialdehyde (MDA) determination using different combination of the words “elisa kits manufacturers in China”, but only the following: https://www.made-in-china.com/productdirectory.do?word=Elisa+Kit+for+MDA&file=&searchType=0&subaction=hunt&style=b&mode=and&code=0&comProvince=nolimit&order=0&isOpenCorrection=1&org=top without details.

Response 1: We have added the details (such as the method of determination, wavelength for spectrophotometry and briefly describe the test process and measurement sensitivity, etc.) in line 136-155.

Point 2. On the other hand, there are a lot of methods of determination of the oxidative stress indicators. For example, there are at least three usual methods of the malondialdehyde (MDA) determination: traditional spectrophotometric determination of the malondialdehyde-thiobarbituric acid (MDA-TBA) complex (method A) and the two HPLC separation methods of the MDA-TBA (method B) and MDA-dinitrophenylhydrazine (MDA-DNPH) adduct (method C). The results of such measurements are more available in the literature as the measurements with “ELISA Kits”. It will be useful, if the authors will explain (in the part “4. Discussion”) the advantages of the of the oxidative stress indicators measurements with “ELISA Kits” in comparison to the usual methods of determination of these indicators, i.e. malondialdehyde (MDA) and others.

Response 2: We have added the details about the advantages of “ELISA Kits” in line 440-449. The four basic formats are direct, indirect, sandwich, and competitive ELISAs. In this study, competitive ELISA were used to detect oxidative stress indexes, utilizes a mixture of antibody-antigen and free antibody in liquid phase to interact with plate-immobilized antigens.

Point 3. Lines 369-370. Please, correct the sentence “This is basically consistent with the results of this study, indicating that acidifiers have a certain effect on reducing diarrhea in weaned piglets.”  (Lines 369-370 in the part “4. Discussion”), because there are no pronounced effects on reducing diarrhea in weaned piglets (see Table 2 on line 203). In addition, the authors wrote “There was no significant difference in diarrhea rate between the four groups during the experimental period (p > 0.05).” (Lines 195-196).

Response 3: Thanks a lot for the critical observations and constructive suggestion. We have correct the sentence about there were no significant effects on reducing diarrhea in weaned piglets. In addition, we also discussed the reasons why the addition of acidifier in drinking water did not improve the diarrhea rate. (Lines 409-416)

Point 4. It will be useful, if the authors will explain (in the parts “3.4. Oxidative Stress Level” or “4. Discussion”) the data concerning the oxidative stress indicators (presented in the Table 4, lines 253-260). For example, there are extremely low and high GSH data of the A2 group (at day 18 and 35, respectively) as compared to the groups A1 and A3 (Table 4). In contrast, there are very high and relatively low T-AOC data of the A2 and A3 groups (at day 18 and 35, respectively) as compared to the groups A1 (Table 4), etc.

Response 4: Thank for this precise observation. We carefully examined the original data and found that when we measured the indicators of oxidative stress at two time points (on day 18 and day 35 respectively), The main possible reason is that individuals have different performance on weaning stress. The weanling stress affects the individual's disease state and immune level. For example, post-weaning diarrhea seriously slows down the growth rate of piglets. Some pigs even die, but some have good adaptability. Individual differences are relatively large in pigs. In addition, improper operation also might affect the results of the experiment. The main reason is that individuals have different performance on weaning stress. As a result, the measurement data (such as GSH and T-AOC) of some individuals was very low or very high at two time points, affecting the overall mean value.

Point 5. There are the interesting data presented at the Figure 5 (Lines 331-335), but there are insufficient explanations of these correlations (between the differential microbial species and measured parameters). It is especially important because of the controversial correlations: strong (weak) or positive (negative) in the case of the similar oxidative stress indicators and the differential microbial species.

Response 5: Thanks a lot for the constructive suggestion. In our revised manuscript, partial correlation analyses were used while accounting for the effects of one or more other variables. We have provide a fully explained the correlation between different microbial species and oxidative stress indicators, and discussed the correlation in detail. (Lines 513-545).

Point 6. Moderate editing of English language and style required. For example, lines 98-100. “There were 4 replicates (pen) in each group, and 25 piglets per pen. The concentration of acidifier is acidifier: water = 1:500, used for 8 h every day, free to drink”.

Please, consider my editing: “There were 4 replicates (pen) in each group, and 25 piglets per pen; used for 8 h every day; free to drink. The ratio of acidifier: water = 1:500 was used in this study”.

Response 6: Thank a lot for this important comments. We totally agree with your suggestion! We have revised it according to your suggestion. (Lines 99-100).

Reviewer 2 Report

Authors have to clarify what statistical analysis they applied and on what amount of samples. It seems to understand on 16 animals for each drinking water group (control and A1,2,3), but in the S1 file it seems to be 6.

In addition, as requested by the journal, the Tables have to be self-explaining; in the Tables 3 and 4 there are many acronims without their meaning.

The rationale of the article is based on previously stated knowledge, so the 

large amount of work seems to be exccessive and the results expected.

Are the parameters examined correlated or influenced among them?

Author Response

Dear Reviewer 2,

Thank you very much for your constructive comments! We are appreciated that the Editor-in-Chief and you gave us an opportunity to revise our manuscript (antioxidants-1664915).

In the revised manuscript, we have meticulously addressed all the comments raised by the reviewers and associate editor. Our response to the comments was listed point-by-point. All the significant changes in the revised manuscript have been highlighted in red.

Point 1: Authors have to clarify what statistical analysis they applied and on what amounts of samples. It seems to understand on 16 animals for each drinking water group (control and A1,2,3), but in the S1 file it seems to be 6.

Response 1: Thank you for your comments and constructive suggestions! We have clarified the amount of sample used to detect the indicators of blood routine, oxidative stress indicators, and microbial sequencing and analysis in line 120, 130, and 163, respectively. In this experiment, a total of 400 weaned pigs at 28 d of age were randomly divided into 4 groups. On d 18 and 35 of experimental period, 64 pigs (4 pigs per pen) were randomly selected to collect blood. The blood was then divided equally into two parts: one for CBC test (n = 128) and the other for determination of oxidative stress indicators (n = 128). On d 18 and 35 of the experiment period, 24 pigs (6 pigs per group) were randomly selected to collect fresh feces from rectum. The feces samples (n = 48) were immediately frozen in liquid nitrogen until DNA extraction.

Point 2: In addition, as requested by the journal, the Tables have to be self-explaining; in the Tables 3 and 4 there are many acronyms without their meaning.

Response 2: Thank you for your precise observation and constructive suggestions! We have explained the abbreviations in tables 3 and 4. (Line 257-262, and Line 285-286, respectively).

Point 3: The rationale of the article is based on previously stated knowledge, so the large amount of work seems to be excessive and the results expected. Are the parameters examined correlated or influenced among them?

Response 3: Thank you for your comments and constructive suggestions! It is accepted that organic acids are considered to be one of the most promising alternatives to antibiotics. However, there is no information on comparing different ratios of acidifiers and to find out which type of acidifiers is more suitable to be added to the drinking water of weaned piglets as an antibiotic substitute. In addition, adding acidifier to drinking water instead of to the feed seems to have unique advantages. Due to the type and dose of dietary organic acids, coating or mixing mode, dietary composition and other factors, the response of dietary organic acids to weaned pigs is different. Compared with supplementation in feed, drinking water supplementation with acidifiers is more convenient and easier to control the dosage in pig farms. Also, the acidifiers can disinfect drinking water and inhibit pathogenic bacteria. We have added details about the innovation of our study and different concerns from others' research in the discussion. (Line 379-384). To further examine whether the parameters examined correlated or influenced among them, we conduct partial correlation analyses between the acidifier-induced microbial alteration and growth performance, antioxidant capacity index. More details are shown in lines 353-360 and 513-545.

Reviewer 3 Report

Review of the manuscript ID 1664915 „Drinking water supplementation with acidifiers improves growth performance with potential regulation of oxidative stress, immunity, and gastrointestinal microbiota diversity of weaned piglets”. ANTIOXIDANTS

The research concerns the issue of health and production indicators of piglets, which is very important for practice. The problem of piglet susceptibility to stress is particularly evident during the piglet weaning period. Any research looking for substitutes for feed antibiotics is absolutely necessary.

L137: “A total of 24 pigs were randomly selected from each pen (3 pigs per pen)” – However, probably it should be: „A total of 24 pigs were randomly selected from each group (3 pigs per pen)”?

L196-197: „…the survival rate of weaned pigs in the A2 group was greater than that of the control group ( p = 0.02)”. However, Table 2 shows p-level for survival rate p=0.107 (not p=0.02), what means there is no any significant difference among groups. Please, chack this statistical analysis once more.

L351-352: „…(containing formic, acetic and propionic acids) with or without propionic acid in piglets”. However it probably should be: …. with or without lactic acid…?

Table 3 and 4: Please, separate the results for the CBC parameters on the 18th day of the experiment and the 35th day. Day 18 results may be at the top of the table (top rows) and the 35 day at the bottom of the table. The current version of table makes it impossible to analyze and compare the results.

Table 3: The abbreviations of blood parameters tested in CBC must be described - either under the table or in the methodology chapter.

Results chapter: Please introduce a uniform system for citation p-level values in the text: either give the exact value from the table (p = 0.865) or use p ≤0.05 and p>0.05, for significant and insignificant differences, respectively.

Discussion chapter:

The Authors measured the blood indices and fecal microbiota status at two dates of piglet rearing: day 18 and day 35. Unfortunately, the time factor does not appear in the Results (showing if the statistical differences exist or not) and has not been discussed. I suggest carrying out an appropriate statistical analysis that will show the effect of time (age of piglets) on the level of the examined indicators. Then, please discuss these results.

L355-358: „…the ADG of piglets in the A1 group was greater than that in the control group, which may be the organic acidifier can effectively stimulate pepsin secretion and activate pepsin activity, so as to improve the utilization rate of feed and make the weight gain of weaned piglets significant”. Your explanation is not quite logical. Acidifier A1 improved the ADG and you explain – because it i san organic acidifier. However, the A3 and A3 are also the organic acidifiers, but causing not any beneficial effects. So?

L366: „Interestingly, the acidifier of the A2 group significantly improved the survival rate of piglets”. I can not see it in Table 2. Table 2 shows p-value at 0.107 for survival rate.

The manuscript needs a major revision.

Author Response

Dear Reviewer 3,

Thank you very much for your constructive comments! We are appreciated that the Editor-in-Chief and you gave us an opportunity to revise our manuscript (antioxidants-1664915).

In the revised manuscript, we have meticulously addressed all the comments raised by the reviewers and associate editor. Our response to the comments was listed point-by-point. All the significant changes in the revised manuscript have been highlighted in red.

Point 1: L137: “A total of 24 pigs were randomly selected from each pen (3 pigs per pen)” – However, probably it should be: „A total of 24 pigs were randomly selected from each group (3 pigs per pen)”?

Response 1: Thank you for your precise observation and constructive suggestions! On d 18 and 35 of the experiment period, 24 pigs (6 pigs per group) were randomly selected to collect fresh feces from rectum. According to your suggestion, we also clarified the amount of sample used to detect the indicators of blood routine, oxidative stress indicators, and microbial sequencing and analysis, respectively.

Point 2: L196-197: „…the survival rate of weaned pigs in the A2 group was greater than that of the control group (p = 0.02)”. However, Table 2 shows p-level for survival rate p=0.107 (not p=0.02), what means there is no any significant difference among groups. Please, check this statistical analysis once more.

Response 2: Thank you for your constructive suggestions! In our study, the diarrhea and survival rate of piglets were analyzed using GLIMMIX procedure of SAS 9.4. Interestingly, we found that acidifier treatment had no significant effect on the survival rate of pigs (p=0.107) in the GLIMMIX model analysis, but there was significant improvement (p = 0.02) on the survival rate of pigs between the A2 group and the control group in the Multiple Comparisons. We have revised this paragraph according to your suggestions.

Point 3: L351-352: (containing formic, acetic and propionic acids) with or without propionic acid in piglets”. However, it probably should be: …. with or without lactic acid…?

Response 3: Thank you for your precise observation! We totally agree with your suggestion. I have revised this sentence.

Point 4: Table 3 and 4: Please, separate the results for the CBC parameters on the 18th day of the experiment and the 35th day. Day 18 results may be at the top of the table (top rows) and the 35 day at the bottom of the table. The current version of table makes it impossible to analyze and compare the results.

Response 4: Thank you for your important suggestions! We have separated the results for the CBC parameters into two parts. The results of day 18 were at the top of the table and the results of day 35 were at the bottom of the table. (Lines 255-256, 283-284, separately).

Point 5: Table 3: The abbreviations of blood parameters tested in CBC must be described - either under the table or in the methodology chapter.

Response 5: Thank you for your comments and constructive suggestions! We have described the abbreviations of blood parameters tested in CBC in Table 3. (Lines 257-262).

Point 6: Results chapter: Please introduce a uniform system for citation p-level values in the text: either give the exact value from the table (p = 0.865) or use p ≤0.05 and p>0.05, for significant and insignificant differences, respectively.

Response 6: Thank you for your comments and constructive suggestions! We have introduced a uniform system for citation p-level values in the text, use p < 0.05 to indicate significant difference and p > 0.05 indicate insignificant differences, 0.05 < p < 0.10 indicate a tendency of difference. The exact p value is uniformly used to indicate whether the difference is statistically significant in the tables, which is generally used in published papers. P-values are reported to three significant digits, except when the p-value is less than 0.001; for p-values smaller than 0.001, it is reported as ‘p < 0.001’. (Lines 193-194).

Discussion chapter:

Point 7: The Authors measured the blood indices and fecal microbiota status at two dates of piglet rearing: day 18 and day 35. Unfortunately, the time factor does not appear in the Results (showing if the statistical differences exist or not) and has not been discussed. I suggest carrying out an appropriate statistical analysis that will show the effect of time (age of piglets) on the level of the examined indicators. Then, please discuss these results.

Response 7: Thank you for your comments and constructive suggestions! According to your suggestion, we put the time factor into our model and re conducted statistical analysis to show the impact of time (age of piglets) on the level of examined indicators. In our study, Data of growth performances, CBC test and oxidative stress indicators were analyzed using GLIMMIX procedure of SAS 9.4 with treatment, gender, time and their interaction as fixed effects, pen as a random effect.

Point 8: L355-358: „…the ADG of piglets in the A1 group was greater than that in the control group, which may be the organic acidifier can effectively stimulate pepsin secretion and activate pepsin activity, so as to improve the utilization rate of feed and make the weight gain of weaned piglets significant”. Your explanation is not quite logical. Acidifier A1 improved the ADG and you explain – because it is an organic acidifier. However, the A3 and A3 are also the organic acidifiers, but causing not any beneficial effects. So?

Response 8: Thanks a lot for your comments! As your suggested, we have discussed the possible reasons why the ADG and ADFI of A1 group (contain lactic acid) are greater than that of the control group, while there is no significant difference between the A2 or A3 group and the control group. In brief, lactic acid combined with compound organic acids containing formic acid, acetic acid and propionic acid may be more helpful to improve the growth performance of pigs. More details are shown in lines 389-402.

Point 9: L366: „Interestingly, the acidifier of the A2 group significantly improved the survival rate of piglets”. I can not see it in Table 2. Table 2 shows p-value at 0.107 for survival rate.

Response 9: Thank you for your comments and precise observation! We have answered this question above. In our study, the diarrhea and survival rate of piglets were analyzed using GLIMMIX procedure of SAS 9.4. Interestingly, we found that acidifier treatment had no significant effect on the survival rate of pigs (p = 0.107) in the GLIMMIX model analysis, but there was significant improvement (p = 0.02) on the survival rate of pigs between the A2 group and the control group in the Multiple Comparisons. (Lines 416-424).

Round 2

Reviewer 1 Report

Dear authors,

The authors made the significant improvements in the corrected manuscript (antioxidants-1664915). I appreciate the corrections in the “Abstract”, but propose to changes the sentence at lines 29-31 (“Subdoligranulum was significantly positive correlation with ADG and ADFI, and Ruminococca-ceae_UCG-005 had a significant positive correlation with serum antioxidant indicators.) as following: “The microbial species Subdoligranulum or Ruminococcaceae_UCG-005 had significantly positive correlations with ADG and ADFI or with serum antioxidant indicators, respectively.”   

All other corrections are fine. The corrected manuscript can be accepted in the present form.  

Reviewer 3 Report

The Authors significantly improved the quality of their manuscript based on suggestions from the Reviewers. Please, accept the text for publication.